# Exploring the Manufacturing Process of a Renaissance Breach Pike

Paolomarco Merico [1], Michela Faccoli [1,*], Roberto Gotti [2] and Giovanna Cornacchia [1]

1 Department of Mechanical and Industrial Engineering, University of Brescia, Via Branze 38, 25123 Brescia, Italy; paolomarco.merico@unibs.it (P.M.); giovanna.cornacchia@unibs.it (G.C.)
2 The Martial Arts Museum, Via Garibaldi 3, 25082 Brescia, Italy; roberto@dismastrading.it
* Correspondence: michela.faccoli@unibs.it

**Abstract:** An archaeometallurgical study of a Renaissance breach pike was performed to elucidate its manufacturing process. Optical microscopy observations and microhardness measurements indicated that the breach pike was forged starting from a heterogeneous steel lump. The microstructural features were compatible with post-forging air cooling. The chemistry of a large set of nonmetallic inclusions was investigated by scanning electron microscopy coupled with X-ray dispersive spectroscopy. Compositional data were analyzed by multivariate statistics to distinguish smelting-related slag inclusions. A logistic regression model indicated that the steel was probably produced by the direct method. The liquidus temperatures of the slag inclusions indicated maximum smelting temperatures in the range of 1200 °C to 1300 °C. A thermodynamic-based model was adopted to estimate the average smelting conditions in terms of temperature and oxygen chemical potential and investigate the disequilibrium of slag inclusion–metal systems. For low-disequilibrium systems, the computed temperature values range between 1095 °C and 1118 °C, while the oxygen chemical potentials ($\mu_{O2}$) span from $-442$ to $-374$ kJ/mol.

**Keywords:** archaeometallurgy; breach pike; scanning electron microscopy; inclusion analysis; thermodynamics

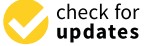

## 1. Introduction

This archaeometallurgical study focused on a specific type of awl-like hafted weapon known as a breach pike. It is essentially composed of a thin, long needle with a square cross-section, mounted on a wooden shaft, and often fixed to it by languets. Moreover, a typical nodus is sometimes positioned between the spike base and the conical socket of the shaft [1]. Based on iconographic sources, it is widely accepted that its initial adoption dates back to at least the 13th century and remained in use until the 16th century (Figure 1) [2].

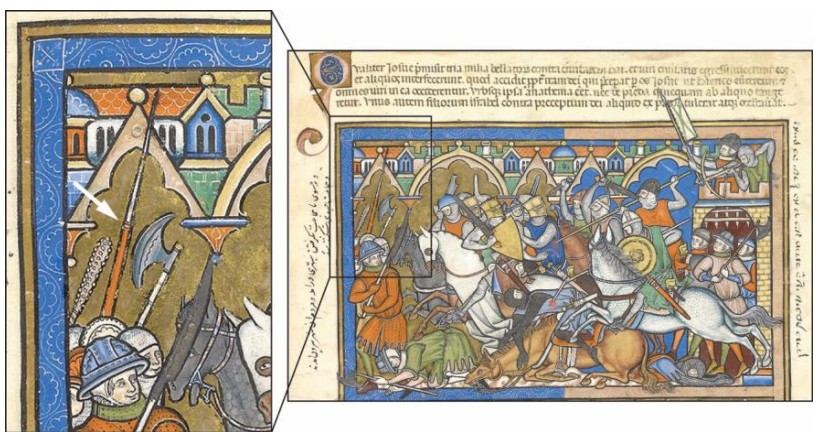

**Figure 1.** Breach Pike (pointed by the white arrow) depicted in a 13th-century miniature (Pierpont Morgan Library New York, Maciejowski Bible, MS M.638, fol. 10r).

Similarly to other staff weapons such as the halberd and spear, the breach pike reached its peak of popularity, especially in Austria and Germany, during the 15th century with the resurgence of professional infantry forces [3]. It was primarily used by the infantry as a thrusting weapon [2]. According to di Carpegna [4], this type of weapon was commonly used to defend breaches opened in the walls or gates of cities and forts during sieges. Moreover, it has been hypothesized that the spread of breach pikes may have been linked to the introduction of plate armors, against which they were highly effective [3]. As amour-piercing weapons, breach pikes needed to possess specific mechanical features, such as toughness, to reduce the likelihood of rupture. In fact, after being damaged in battle, a bent breach pike could be repaired and was still more effective than a broken one [5]. It is generally assumed that the fabrication of breach pikes did not require the use of high-quality iron and steel or complex forging operations [5]. However, the manufacturing technology involved in the breach pike fabrication process has not been thoroughly investigated so far. To the authors' knowledge, the only metallurgical analysis results currently available in the literature refer to a breach pike from the late medieval archaeological site of Lubrza Castrum (Poland) [5]. This study has ascertained that the breach pike was forged by combing steel and phosphoric iron without executing any post-forging heat treatment [5].

Given this scenario, this study aims to provide new deep insights into the iron-making technology and forging techniques employed in the production of breach pikes. To achieve this goal, a breach pike donated from the private collection of the "The Martial Arts Museum" in Botticino (Brescia, northern Italy) was carefully analyzed following a state-of-the-art methodology based on a metallography and slag inclusions (SI) analysis [6–11].

## 2. Materials and Methods

### 2.1. The Breach Pike: Preliminary Description and Macro-Observations

The breach pike is characterized by a conical socket and a quadrangular spike, the terminal portion of which is significantly bent and the tip is missing, likely due to a damaging phenomenon (e.g., a violent impact sustained during battle) or an intentional re-shaping process (Figure 2a). Moreover, the spike profile appears slightly buckled (Figure 2b). A nodus between two grooves is positioned in the transition zone from the socket to the beveled spike base (Figure 2b). For the majority of its length, the socket exhibits a seam that is not completely sealed (Figure 2b). The total lengths of the breach pike (excluding the absent tip), its socket, and its spike are 380 mm, 98 mm, and 265 mm, respectively. The cone-shaped socket has an external diameter that diminishes uniformly from 29 mm to 14 mm, whereas the spike width tapers from 15 mm to 5 mm at the broken end. Furthermore, two nail holes were punched near the socket base to reinforce the joining of the break pike to the wooden shaft through nailing. To carry out metallographic and SI analyses, a single transverse section (S1) was extracted from the socket, whereas a longitudinal section (S2) and two cross-sectional segments (S3 and S4) were taken from the socket–spike transition zone and spike, respectively. The sampling strategy is detailed in Figure 2c.

### 2.2. Experimental Methods

In this research work, the experimental protocol specifically tailored by the authors for the archeometallurgical examination of other historical ferrous artefacts was applied [6–9]. In particular, each section was prepared following a standard metallographic procedure consisting of grinding with SiC abrasive papers (grades from 600 to 1200) and polishing with polycrystalline diamond pastes (3 μm and 1 μm). A preliminary metallographic investigation was carried out on the as-polished sections by means of an optical microscope (Leica DMI 5000M, Wetzlar, Germany) to characterize the SI morphology and position. The material's microstructure was investigated after chemical etching by immersion in 1 pct Nital solution (1 vol pct $HNO_3$ in ethanol). Vickers microhardness measurements were performed using a Mitutoyo HM-200 microhardness tester, imposing a 300 g load for a 15 s dwell time on the etched section. A representative sample of SI was selected from longitudinal S2 and transverse S3 cross-sections after repolishing, and compositional

data were collected semiquantitatively through area acquisition via Energy-Dispersive X-ray Spectroscopy (EDS) coupled with a scanning electron microscope (LEO EVO 40 XVP, Carl Zeiss AG, Milan, Italy). The chemical analysis of multi-phase SIs was performed by scanning all SI surfaces. The chemical data are given in terms of the following stoichiometric oxides: $Na_2O$, $MgO$, $Al_2O_3$, $SiO_2$, $K_2O$, $CaO$, $TiO_2$, $MnO$, $FeO$, $SO_2$, and $P_2O_5$. The SI mineralogy was investigated by SEM in backscattered-electron mode (BSD). The carbon content in the metal matrix adjacent to each analyzed SI was estimated using the inverse lever rule [12]. This calculation was based on the volume fractions of ferrite and pearlite measured by the Image J 1.54a software environment for image analysis [13].

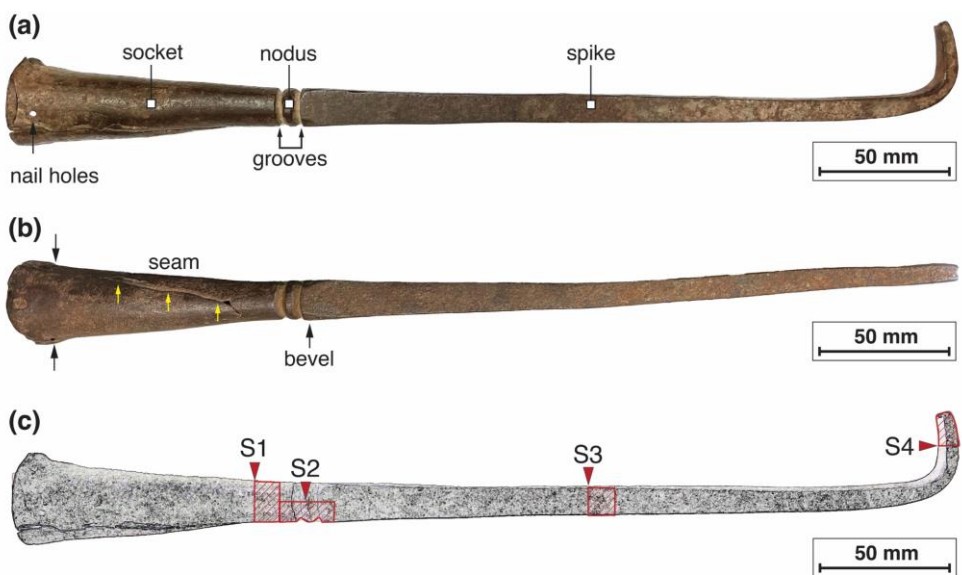

**Figure 2.** (**a**,**b**) Photographic images of the breach pike from two different angles with key terminology and (**c**) sampling methodology.

### 2.3. Slag Inclusion Classification and Data Treatment

In ancient direct and indirect iron smelting processes, the inefficient separation between molten metallurgical slag and solid ferrous alloys promoted the incorporation of a large number of smelting-related slag inclusions (SI) into the metal matrix [14]. In fact, direct iron ore reduction within the bloomery furnace and the indirect production of iron and steel via the decarburation of cast iron occurred below the metal liquidus temperature [14]. Moreover, slag inclusions can also be generated during the smithing stage. For example, iron oxide scale residues and fayalitic slag remnants originating from interactions between flux and iron oxide scale during hammer-welding can remain embedded at welding interfaces as forging SI. The analysis of slag inclusions is a powerful tool to investigate the iron-making method and the forging strategy adopted [10,15]. Specifically, smelting and forging-related SI potentially encapsulate a chemical signature and thermochemical information associated with smelting and forging systems, respectively [16]. To achieve this goal, the classification strategies suggested by Dillmann and L'Héritier [16] and Charlton et al. [17] were applied to differentiate smelting and forging SI. These methods describe SI as multicomponent oxide systems for which two main oxide categories can be conveniently distinguished: reduced compounds (RCs) and non-reduced compounds (NRCs). Remarkably, non-reduced compounds (NRCs) such as $Al_2O_3$, $K_2O$, $MgO$, and $CaO$ serve as the key chemical variables for identifying slag inclusion (SI) groups associated with distinct smelting and forging systems. In contrast, the levels of reduced compounds (RCs), such as $FeO$ and $P_2O_5$, are highly influenced by redox condition variations. Consequently, RCs generally feature a high dispersion even within SI groups originating from the same technological system. According to Dillmann and L'Héritier [16], SI derived from the same smelting system (marked by the specific iron ore, lining, charcoal, and fluxes used) exhibit

similar NRC ratios. On the other hand, a higher $SiO_2$–$Al_2O_3$ ratio is a peculiar feature of forging-related SI, as silica-rich smithing flux was often used [16]. Before performing the PCA, the magnitude and variance of the NRCs were equalized by pre-processing the SI compositional dataset through a centered log–ratio transformation (CLR), as expressed by Equation (1):

$$Xi_{NRC} = \log(Ei_{NRC}) - g(\log E_{NRC}) \tag{1}$$

where $Xi_{NRC}$ is the $i_{th}$ transformed value for each well-quantified NRC (i.e., CaO, MnO, $Al_2O_3$, and $SiO_2$), $Ei_{NRC}$ is the amount of the $i_{th}$ NRC, and $g(\log E_{NRC})$ is the geometrical mean of the logged NRCs. A HCA based on the Ward method with Euclidean distance was carried out for the SI clustering. The data treatment was performed using R 3.6.1, Excel, and OriginLab 9.9 software environment for statistical computing and graphics [18].

*2.4. Thermodynamic Modelling of Slag Inclusion–Metal Systems*

The thermodynamic-based model developed and described in detail by the authors in previous research work [6] was employed to estimate the temperature (T) and oxygen chemical potential ($\mu_{O2}$) of the gas phase equilibrated with each SI–metal system (i.e., SI along with the surrounding metallic phase). These thermochemical parameters can provide an indication of the iron smelting working conditions. Notably, this model relies on the assumption that thermodynamic equilibrium was locally attained. In this respect, it is worth noting that an inverse correlation between the iron oxide concentration in smelting slag inclusions and the carbon content in the metal matrix has been frequently reported for ancient heterogeneous ferrous alloys produced through direct and indirect iron-making processes [19]. This phenomenon indicates that some SI–metal systems likely approach thermodynamic equilibrium [6,20]. Moreover, it was assumed that the iron oxide content in SI and the carbon concentration in the metal matrix are mainly governed by the gas phase generated from the reaction of air oxygen with charcoal in bloomery or fining furnaces. The gas phase was modelled as a $CO/CO_2/N_2$ mixture with small traces of oxygen at a pressure of 1 bar (0.1 MPa). The temperature (T) and oxygen chemical potential ($\mu_{O2}$) were computed by solving Equation (2) for each SI–metal system:

$$\mu_{O2} ([C], T, [w_M]) = \mu_{O2} ((FeO), T, (w_S)) \tag{2}$$

where [C] and (FeO) are the weight percentages (wt%) of carbon and iron oxide in the austenitic metal ($\gamma$) and liquid (or glass) SI phases, respectively. In addition, $[w_M]$ and $(w_S)$ correspond to the weight percentages (wt%) of other elements and constituents in the metal and SI phases, respectively. The equilibrium condition for each SI–metal system graphically corresponds to the intersection point of the iso-C and iso-FeO concentration curves in the oxygen chemical potential space. On the other hand, some SI–metal systems do not follow the FeO-C inverse correlation and, therefore, exhibit a significant degree of thermodynamic disequilibrium (e.g., high-FeO and low-FeO SI encased within high-C and low-C steel, respectively). It has been speculated that the disequilibrium level of SI–metal systems can be exacerbated by post-smelting thermochemical treatments such as carburization and decarburization [19]. Moreover, the SI–metal systems generated by the incorporation of forging SI at hammer welding joints are likely marked by higher disequilibrium than those that originated from the iron smelting process. Consequently, the characterization of the SI–metal systems' disequilibrium potentially provides the opportunity to expand the SI classification and obtain valuable technological information concerning the breach pike manufacturing process. To highlight and identify both low- and high-disequilibrium systems, the thermodynamic model was applied to all the SI–metal systems. The high level of positive or negative deviation from the central tendency in the frequency distribution of the estimated equilibrium temperatures was used as a selection criterion for high-disequilibrium SI–metal systems. Specifically, significant negative and positive deviations are expected for high-FeO SI surrounded by a high-C metal matrix (labelled "A" in Figure 3) and low-FeO SI surrounded by a low-C metal matrix (labelled "C" in Figure 3). It is

important to remark that, for these latter systems, the model outcomes do not reflect real thermochemical conditions but serve as proxies for assessing the level of disequilibrium.

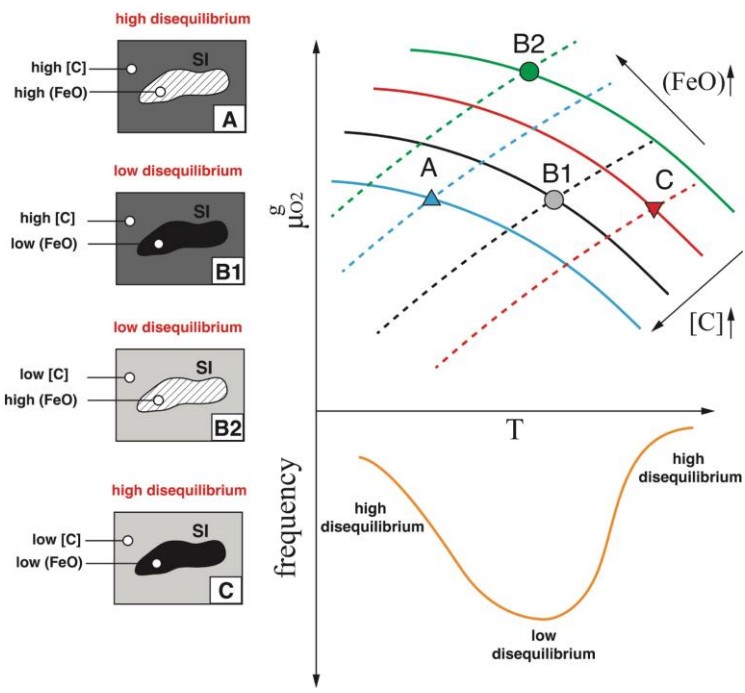

**Figure 3.** Strategy for interpreting the thermodynamic model outcomes in terms of disequilibrium levels of SI–metal systems.

## 3. Results and Discussion

### 3.1. Metallographic Examination and Microhardness Measurements

A detailed metallographic investigation was conducted on all four sections taken from the breach pike socket (S1), socket–spike transition zone (S2), and spike (S3, S4). The examination by light optical microscopy of the as-polished sections S1 (Figure 4a), S2 (Figure 5a), S3 (Figure 6a), and S4 (Figure 6c) revealed a considerable quantity of SI and cracks. The as-polished transverse section S1 featured surface cracks which are partially filled with oxidation products and surrounded by clusters of nonmetallic inclusions (Figure 4b). These cracks may have originated during the socket shaping process through hot working operations. Furthermore, an internal crack was identified within section S1 (Figure 4c), which is likely due to an incomplete hammer-welded seam between the two flaps that were wrapped around a mandrel to form the conical socket. This hypothesis is supported by the previously described imperfect seam on the socket surface (Figure 2b).

The core region of the longitudinal section S2 is characterized by cracks and SI alignments oriented along the main hot plastic deformation direction (Figure 5b). These features likely correspond to the incomplete seam previously observed in section S1. Moreover, discontinuous SI bands were observed near the surface, suggesting that the circular grooves were obtained by post-forging grinding operations (Figure 5c).

Highly scattered SI were detected in the spike transverse sections S3 and S4 (Figure 6a,c). In addition, a slightly distorted SI line extending from the outer to the inner portion of the S3 section was observed (Figure 6c).

Chemical etching with 1 pct Nital of the S1 section highlighted a highly heterogeneous microstructure, suggesting a non-uniform carbon content. As shown in Figure 6a, the majority of the section is characterized by equiaxed ferrite grains (F) (Figure 7b) and ferritic–pearlitic zones (F + P) (Figure 7c). In addition, an elongated and irregularly shaped region is marked by pearlite with minor fractions of idiomorphic, allotriomorphic, and Widmanstätten ferrite (P + F*) (Figure 7d).

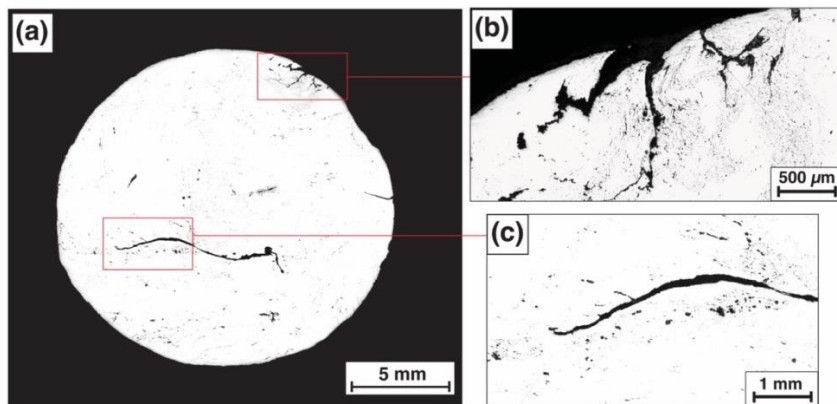

**Figure 4.** Features of as-polished section S1. (**a**) As-polished transverse section S1 (collage of optical micrographs at 50× magnification), (**b**) surface cracks, and (**c**) internal crack.

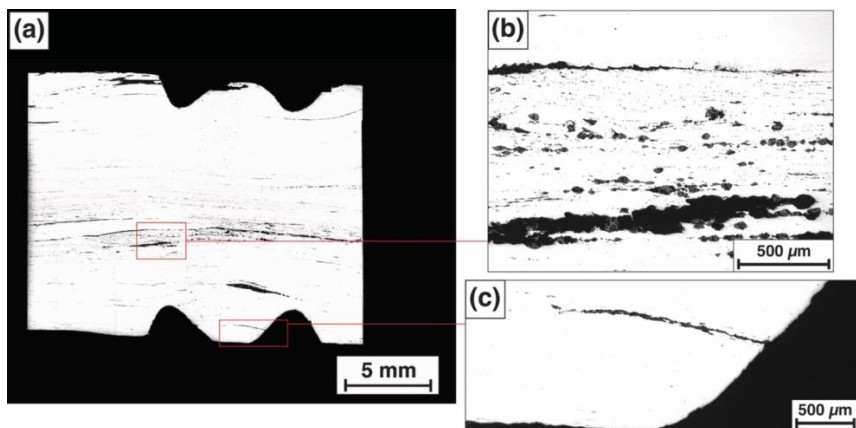

**Figure 5.** Features of as-polished section S2. (**a**) As-polished longitudinal section S2 (collage of optical micrographs at 50× magnification), (**b**) cracks and SI bands, and (**c**) truncated SI line near the outer surface of the groove.

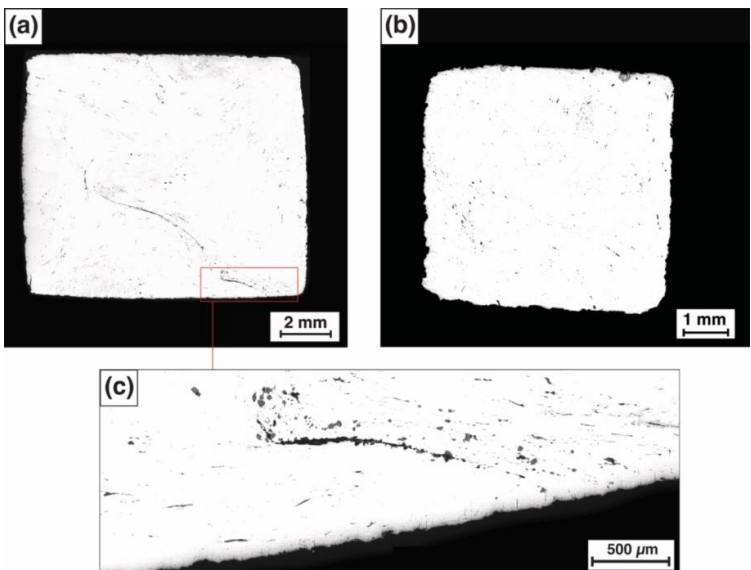

**Figure 6.** Features of as-polished sections S3 and S4. As-polished spike transverse sections (**a**) S3 and (**b**) S4 (collage of optical micrographs at 50× magnification); (**c**) SI cluster near the outer surface of section S3.

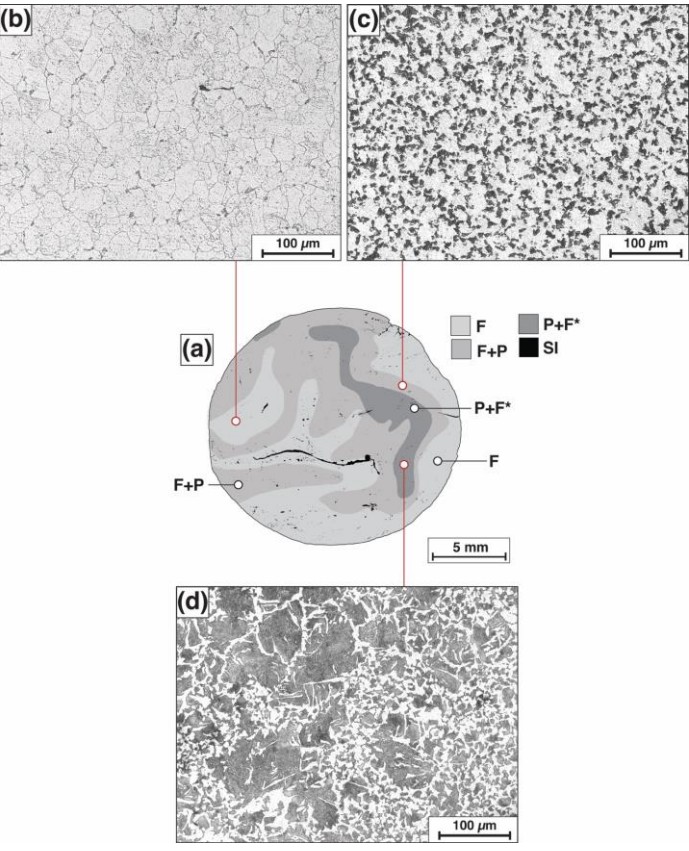

**Figure 7.** Microstructural features of section S1. (**a**) Distribution map of the main microstructural constituents and features observed in the S1 section and (**b**–**d**) some representative optical micrographs at 200× magnification (F = equiaxed ferrite, F* = idiomorphic, allotriomorphic, and Widmanstätten ferrite, P = pearlite, SI = slag inclusions).

Coherently, the longitudinal section S2 is characterized by a multi-layered microstructure (Figure 8a). In particular, the core and outer regions of the section are formed by equiaxed ferrite grains (F) (Figure 8b). Furthermore, the ferritic microstructure is locally characterized by the abnormal growth of some ferrite grains (F**) (Figure 8c). It is worth noting that the coexistence of SI aligned with the plastic deformation directions and equiaxed ferrite grains clearly indicates that a recrystallization phenomenon occurred during material permanence above the recrystallization temperature. In addition, intragranular needle-like carbides were precipitated within the larger ferrite grains (Figure 8c, high-magnification image in Figure 9). In accordance with the observed coarsening of ferrite grains, the precipitation of these intragranular carbides likely occurred during a prolonged annealing of carbon-supersaturated ferrite. The remaining portion of the section is characterized by layers of equiaxed ferritic grains with pearlite traces (F + P) (Figure 8d) and ferritic–pearlitic bands with idiomorphic, allotriomorphic, and Widmanstätten ferrite (P + F*) (Figure 8e).

Consistent with the previous results, the optical examination of the etched section S3 exhibited a highly heterogeneous microstructure consisting of ferrite and pearlite (F + P) (Figure 10a). Similar to what was observed in section S2, the ferritic zones (F**) are featured by coarse grains with intragranular acicular carbides (Figure 10b). The other regions of section S3 are characterized by ferritic–pearlitic zones (F + P) (Figure 10c) surrounding mainly pearlitic regions with a minor amount of idiomorphic, allotriomorphic, and Widmanstätten ferrite (F* + P) (Figure 10d).

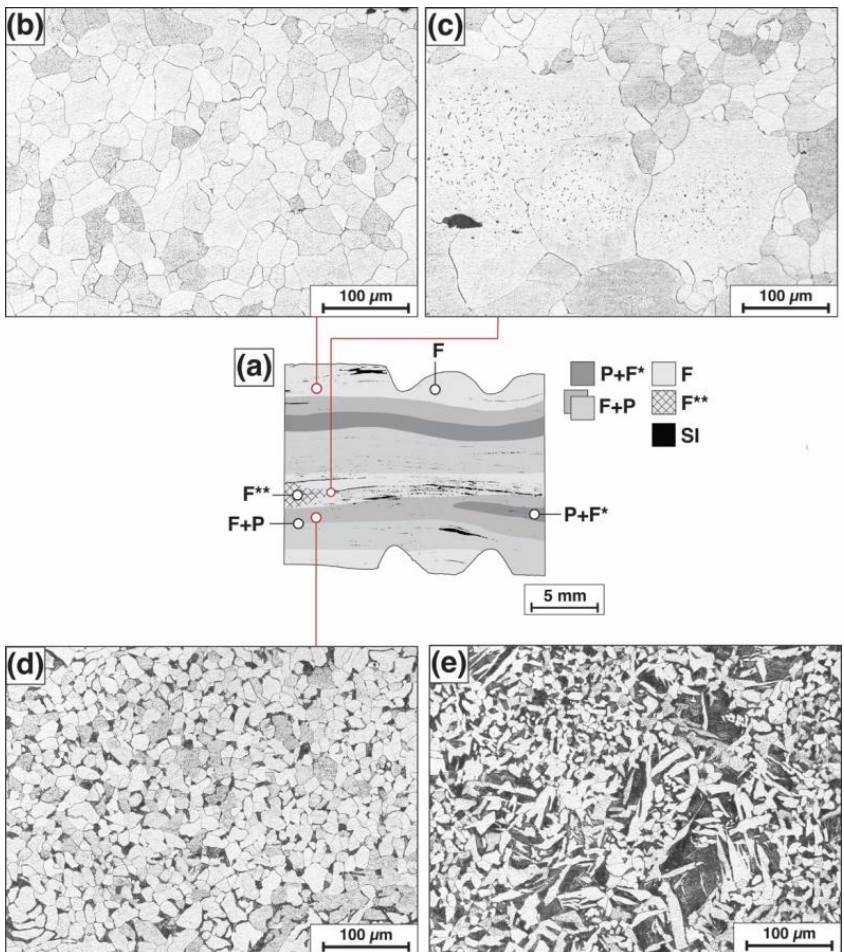

**Figure 8.** Microstructural features of section S2. (**a**) Distribution map of the main microstructural constituents and features observed in the S1 section and (**b**–**e**) some representative optical micrographs at 200× magnification (F = equiaxed ferrite, F* = idiomorphic, allotriomorphic, and Widmanstätten ferrite, P = pearlite, SI = slag inclusions, F** = ferrite characterized by abnormal grain growth).

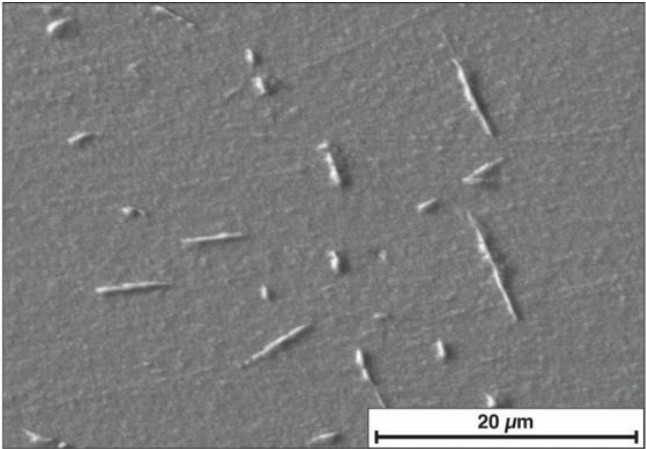

**Figure 9.** Scanning electron microscopy image (SEM-SE) of acicular-shaped carbides precipitated within a ferrite grain.

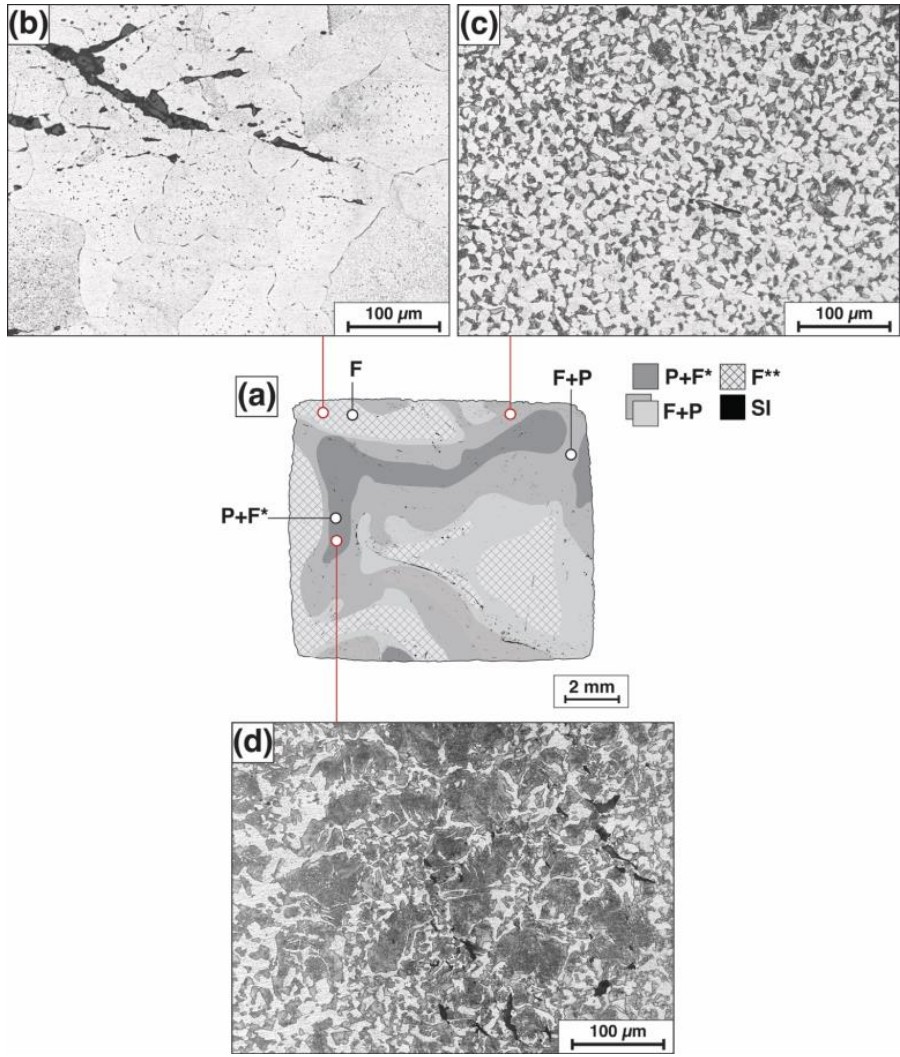

**Figure 10.** Microstructural features of section S3. (**a**) Distribution map of the main microstructural constituents and features observed on the transverse section S3 and (**b**–**d**) some representative optical micrographs at 200× magnification (F = idiomorphic ferrite, F* = idiomorphic, allotriomorphic and Widmanstätten, ferrite, F** = ferrite characterized by abnormal grain growth, P = pearlite, SI = slag inclusions).

The etched transverse section S4 taken from the end portion of the spike exhibited a complex microstructure (Figure 11a). The majority of the section is characterized by a mixture of ferrite–austenite aggregates, often referred to as granular bainite (GB), with large martensite/retained austenite islands (MA) and ferrite (F) (Figure 11b; high-magnification SEM-SE image in Figure 12a) [21,22].

In a restricted zone located near the upper right corner of the section, the steel reached a hypereutectoid composition. In fact, its microstructure is characterized by a proeutectoid cementite network (C) formed at the prior austenite grain boundaries and bainite (B) (Figure 11c; high-magnification SEM-SE image in Figure 12b). Moreover, two regions positioned near the section corners featured a ferritic microstructure with slightly stretched grains (F) (Figure 11d). On the other hand, close to the lower right edge of the section, the ferrite grains exhibited noticeable elongation, likely due to the cold working induced by cold hammering, possibly during reworking operations or as a result of a tip damage event.

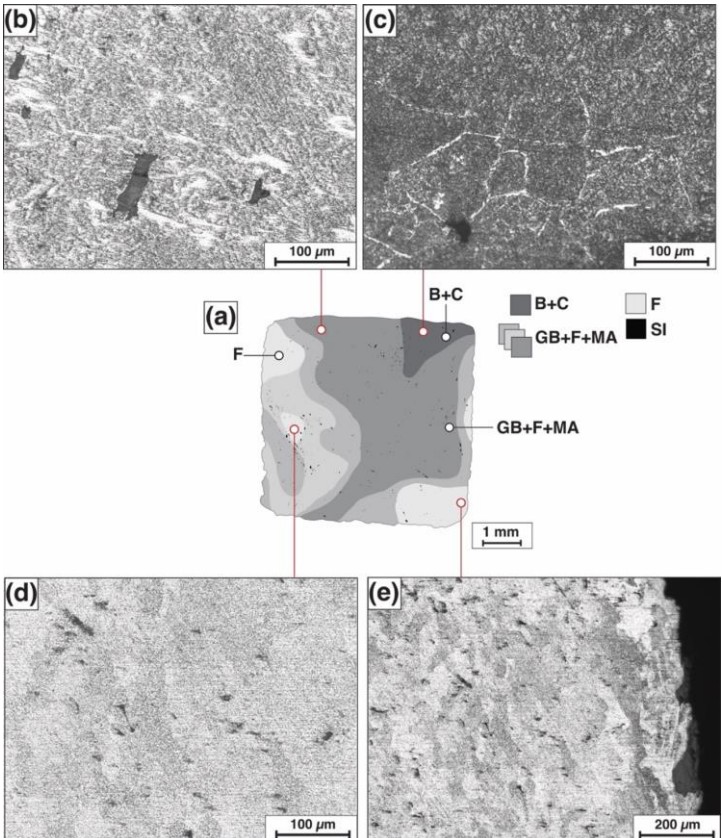

**Figure 11.** Microstructural features of section S4. (**a**) Distribution map of the main microstructural constituents and features observed on the transverse section S4 and (**b–d**) some representative optical micrographs at 200× magnification and (**e**) 100× magnification (F = ferrite, GB = granular bainite, MA = martensite/austenite islands, B = bainite, C = cementite, SI = slag inclusions).

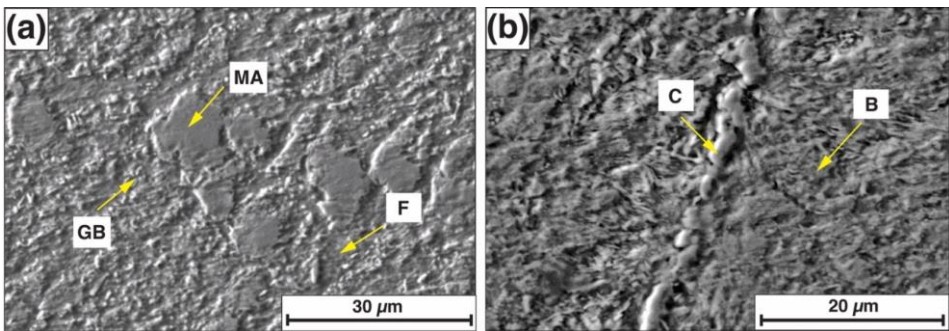

**Figure 12.** Scanning electron microscopy images (SEM-SE) of microstructures observed in section S4. (**a**) Mixture of granular bainite (GB), martensite/retained austenite islands (MA), ferrite (F), and (**b**) bainite (B) along with proeutectoid cementite (C).

Vickers microhardness measurements were performed for all four sections to validate the metallographic results. At least five indentations were carried out for each relevant microstructure. Specifically, in the ferritic zones, the microhardness values vary in the range of 97 to 140 $HV_{0.3}$, whereas a range of 103 to 168 $HV_{0.3}$ was observed in the ferritic–pearlitic regions. On the other hand, in predominantly pearlitic areas featuring idiomorphic, allotriomorphic, and Widmanstätten ferrite, the microhardness values varied from 131 to 179 $HV_{0.3}$. In addition, the mixture of granular bainite, martensite/retained austenite islands, and ferrite featured microhardness values between 200 and 272 $HV_{0.3}$. In particular, the microhardness values of granular bainite and martensite/retained austenite

island constituents are in the ranges 241 to 320 $HV_{0.05}$ and 418 to 485 $HV_{0.05}$, respectively. Moreover, the bainitic microstructure with proeutectoid cementite featured microhardness values varying between 317 $HV_{0.3}$ and 387 $HV_{0.3}$.

Based on the results, it can be concluded that the breach pike was forged by hot hammering. In particular, the breach pike was shaped from a heterogeneous hypoeutectoid steel block. One side of this lump was flattened and coiled around a mandrel to obtain the conical socket of the breach pike. The intermediate region between the socket and the spike was then rounded, possibly using a dedicated swage block. On the other hand, the quadrangular spike was formed by drawing out and pointing operations. After forging, the breach pike was cooled in air, as pointed out by the presence of ferrite and pearlite in sections S1, S2, and S3. Conversely, in accordance with the bainitic microstructure detected in section S4, the region near the spike tip underwent more rapid cooling. This phenomenon can be intentionally promoted to increase the tip hardness, and it may be further exacerbated by the decreasing thickness of the spike toward its tip. However, these hypotheses should be considered with caution as the original microstructure of the pike, especially towards its end, could have been altered during the possible reworking process. In addition, the circular grooves at the socket–spike transition zone were carved by a post-forging material removal process with a specialized tool.

### 3.2. Slag Inclusion Classification

The compositional data of 43 SI were collected via an SEM/EDS analysis (Supplementary Table S1). The results of the multivariate statistical analysis, performed using PCA in conjunction with HCA, are provided in Figure 13. Specifically, two SI clusters were detected in the PC space (Figure 13a) after cutting the dendrogram plot at a height of two (Supplementary Figure S1).

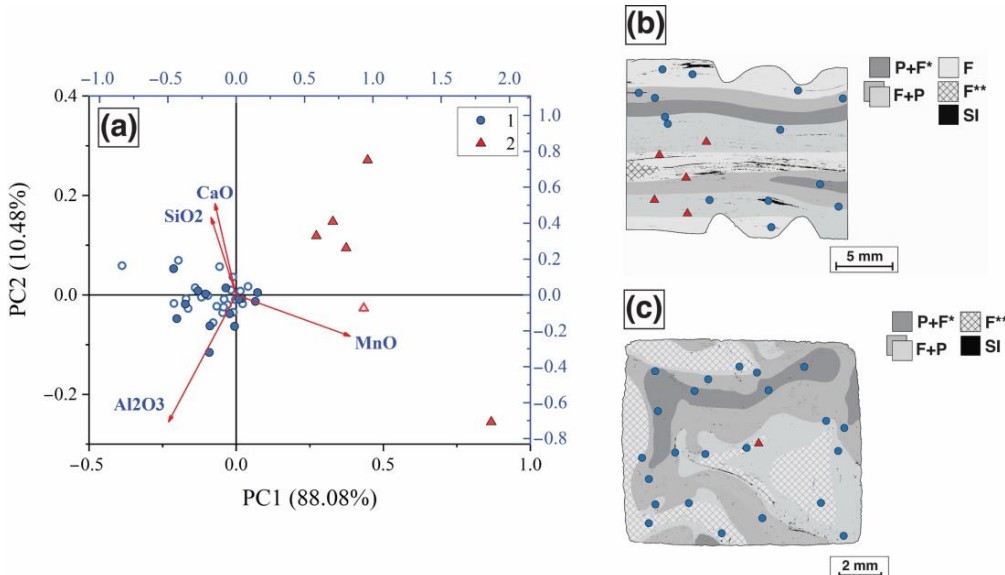

**Figure 13.** Results of multivariate statistical analysis (PCA/HCA). (**a**) Projection of SI chemistry on PC1–PC2 plane (solid and open versions of the same symbol refer to transverse, S2, and longitudinal, S3, sections, respectively) with loading vectors plot (the percentage of variation explained by each PC is indicated in brackets) and location of SI groups on microstructural constituents maps of (**b**) section S2 and (**c**) section S3 (F = idiomorphic ferrite, F* = idiomorphic, allotriomorphic, and Widmanstätten ferrite, F** = ferrite characterized by abnormal grain growth, P = pearlite, SI = slag inclusions).

For each SI group, the area-weighted mean percentages of oxides were calculated using Equation (3):

$$\text{wt\% Oxide}_j^* = \sum_{i=1}^{n_j} \text{wt\% Oxide}_{ij} \times \frac{S_{ij}}{S_{Tj}} \tag{3}$$

where wt% Oxide*$_j$ represents the weighted average percentage of a specific oxide for the *j*th group, wt%Oxide$_{ij}$ denotes the oxide weight percentage for the i*th SI in the *j*th cluster, $S_{ij}$ corresponds to the *i*th SI surface in the *j*th group, and $S_{Tj}$ is the total surface area of the *j*th group. The weighted average percentages of oxides for each SI group are listed in Table 1.

**Table 1.** Average weighted oxide percentages (wt%) for each SI group.

| Group | Na$_2$O | MgO | Al$_2$O$_3$ | SiO$_2$ | K$_2$O | CaO | TiO$_2$ | MnO | FeO | SO$_2$ | P$_2$O$_5$ |
|---|---|---|---|---|---|---|---|---|---|---|---|
| **1** | 0.07 | 4.03 | 6.96 | 36.84 | 2.08 | 11.93 | 0.88 | 18.87 | 17.75 | 0.30 | 0.27 |
| **2** | 0.00 | 4.24 | 1.15 | 12.18 | 0.42 | 4.15 | 0.49 | 22.88 | 53.82 | 0.02 | 0.67 |

It was found that group 1 (blue circles), which includes the majority of the analyzed SI, is mainly located near the origin of the PC1–PC2 plane (Figure 13a). According to Charlton et al. [17], this result supports the hypothesis that group 1 contains smelting-related SI. This assumption is further confirmed by the significant amounts of MnO, which are likely derived from manganiferous iron ore gangue (Table 1). Moreover, the high percentages of Al$_2$O$_3$ and CaO can be attributed to clay and CaO-rich iron ore or lime flux contaminations, respectively. On the other hand, group 2 (red triangles) is positioned in opposition to the Al$_2$O$_3$ loading vector in the PC1–PC2 plane (Figure 13a). Accordingly, this group exhibits decreased concentrations of Al$_2$O$_3$, SiO$_2$, and CaO with respect to group 1, while showing elevated levels of FeO and MnO (Table 1). As previously noted for group 1, the high level of MnO serves as an effective indicator that allows us to classify group 2 as smelting-related. To confirm the SI classification, an FeO–SiO$_2$ (Figure 14a) scatterplot was examined, along with two NRC bivariate scatterplots: SiO$_2$–Al$_2$O$_3$ (Figure 14b) and MnO–SiO$_2$ (Figure 14c). Consistent with the PCA/HCA results, the point clouds for groups 1 and 2 reveal divergent trends attributable to their differing NRC ratios. In particular, group 1 exhibits lower SiO$_2$–Al$_2$O$_3$ and MnO–SiO$_2$ ratios than group 2. It should be noted that group 1 displays a high degree of scatter in the MnO–SiO$_2$ plot, deviating significantly from the expected linear trend. This phenomenon can be explained by a significant variation in MnO in the iron ore, which promotes a higher variation in Mn [23].

The positions of the SI groups on the microstructural constituent maps of sections S2 and S3 are shown in Figure 13b,c. Notably, smelting-related group 1 is located in the external part of section S2. Except for one SI, all the analyzed SI in section S2 belong to group 1. On the other hand, group 2 is preferentially positioned in the mainly ferritic inner portions of sections S2 and S3. To expand the SI characterization, the mineralogy of each SI was investigated by SEM/EDS. Based on their mineralogical features, two main SI types were identified, i.e., two-phase manganese–wüstite–olivine SI (Figure 15a) and single-phase glassy SI (Figure 15b). Example chemical data for the mineralogical phases indicated in Figure 15 are detailed in Table 2. It was found that the majority of SI included in group 2 are predominantly composed of manganese–wüstite with small traces of olivine. These mineralogical characteristics explain the observed chemical diversity of group 2 with respect to group 1 due to the poor compatibility of silicon, aluminum, and calcium with the manganese–wüstite phase.

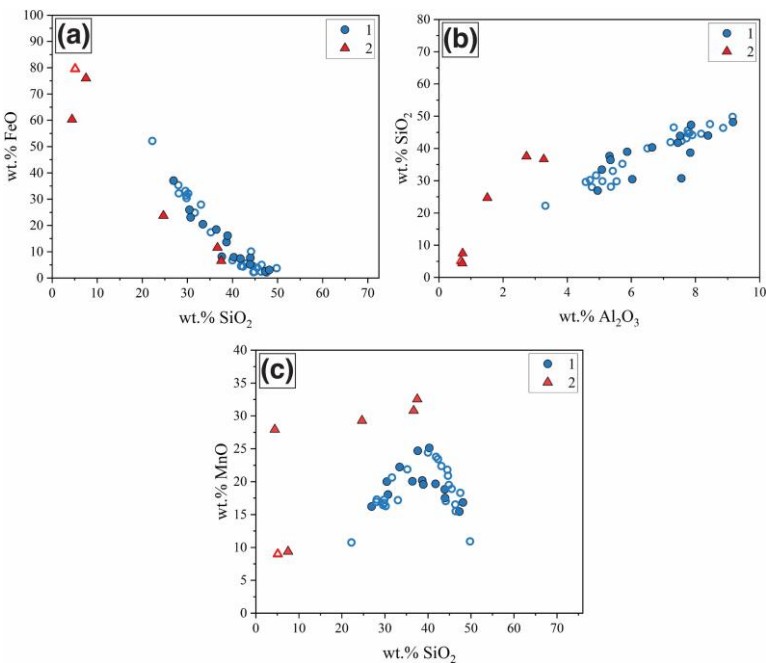

**Figure 14.** Bivariate scatterplots: (**a**) FeO–SiO$_2$, (**b**) SiO$_2$–Al$_2$O$_3$, and (**c**) MnO–SiO$_2$ (open and solid versions of the same symbol refer to S2 and S3 sections, respectively).

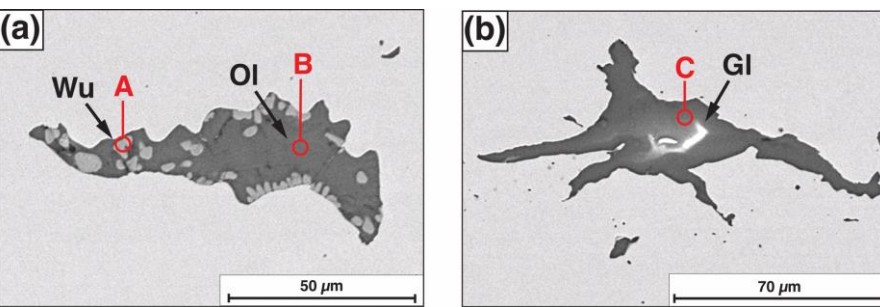

**Figure 15.** SEM micrographs of the two main SI mineralogical types. (**a**) two-phase manganese–wüstite (grey)–olivine (dark grey) SI and (**b**) single-phase glassy SI; Wu: manganese–wüstite; Ol: olivine; Gl: glass. The letters A, B, and C indicate the zones chemically analyzed by SEM/EDS.

**Table 2.** SEM/EDS microchemical analyses (wt%) of the areas highlighted in Figure 15.

|   | Na$_2$O | MgO | Al$_2$O$_3$ | SiO$_2$ | K$_2$O | CaO | TiO$_2$ | MnO | FeO | SO$_2$ | P$_2$O$_5$ |
|---|---|---|---|---|---|---|---|---|---|---|---|
| **A** | 0.00 | 2.60 | 0.00 | 0.00 | 0.00 | 0.00 | 0.00 | 19.46 | 77.93 | 0.00 | 0.00 |
| **B** | 1.04 | 2.61 | 6.84 | 36.77 | 2.23 | 12.64 | 0.00 | 16.13 | 20.51 | 0.00 | 1.22 |
| **C** | 2.44 | 5.53 | 7.82 | 44.89 | 2.29 | 14.51 | 0.73 | 19.50 | 2.31 | 0.00 | 0.00 |

### 3.3. Analysis of the Steel Technological Origin

The logistic regression model suggested by Disser et al. [15] was applied to investigate the technological origin of the ferrous material utilized in the manufacturing of the breach pike. Based on the chemical composition of smelting-related SI, this methodology permits estimating the probability of the breach pike's raw iron/steel being produced through either direct or indirect iron-making routes by comparison with a reference database. According to Disser et al. [15], the average weight oxide percentages of smelting-related SI (Oxide*) were converted into subcompositional ratios (Oxide**) using Equation (4):

$$\text{wt\% Oxide**} = (\text{wt\% Oxide*} \times 100)/(100 - \text{wt\% FeO*}) \tag{4}$$

where the term "Oxide" denotes each oxide involved in the statistical model (i.e., MgO, $Al_2O_3$, $SiO_2$, $P_2O_5$, $K_2O$, CaO, MnO). Given the preprocessed compositional data and the model parameters reported in Table 3 ($\beta^0$, $\beta^{Mg}$, $\beta^{Al}$, $\beta^{Si}$, $\beta^P$, $\beta^K$, $\beta^{Ca}$, $\beta^{Mn}$), the logistic regression model allows us to compute the probability (p) that the ferrous material was obtained by the indirect iron-making method, as shown in Equation (5):

$$\text{Logit(p)} = \log [p/(1-p)] = \beta^0 + \beta^{Mg} [\text{wt.\% MgO**}] + \beta^{Al} [\text{wt.\% Al}_2\text{O}_3\text{**}] + \beta^{Si} [\text{wt.\% SiO}_2\text{**}] + \beta^P [\text{wt.\% P}_2\text{O}_5\text{**}] + \beta^K [\text{wt.\% K}_2\text{O**}] + \beta^{Ca} [\text{wt.\% CaO**}] + \beta^{Mn} [\text{wt.\% MnO **}] \quad (5)$$

**Table 3.** Logistic regression model parameters.

|  | Parameter | Value |
|---|---|---|
| **Intercept** | $\beta^0$ | 5.22 |
| **MgO** | $\beta^{Mg}$ | 0.13 |
| **Al$_2$O$_3$** | $\beta^{Al}$ | −0.95 |
| **SiO$_2$** | $\beta^{Si}$ | 0.007 |
| **P$_2$O$_5$** | $\beta^P$ | 0.16 |
| **K$_2$O** | $\beta^K$ | −0.84 |
| **CaO** | $\beta^{Ca}$ | 0.088 |
| **MnO** | $\beta^{Mn}$ | 0.018 |

In particular, if the computed value of p is either below 0.3 or above 0.7, the iron-making process is predicted to be direct or indirect, respectively. Conversely, due to the existence of an overlapping area for the chemical features of SI derived from direct and indirect smelting processes, the attribution should be labelled 'undetermined' when the value of p lies within the range of 0.3 to 0.7 [15]. In the present research, only the smelting-derived SI of group 1 were considered. The calculated oxide subcompositional ratios for this group are detailed in Table 4.

**Table 4.** Oxide subcompositional ratios for smelting-derived SI of group 1.

| Group | MgO** | Al$_2$O$_3$** | SiO$_2$** | K$_2$O** | CaO** | MnO** | P$_2$O$_5$** |
|---|---|---|---|---|---|---|---|
| 1 | 4.89 | 8.47 | 44.79 | 2.53 | 14.50 | 22.94 | 0.33 |

Applying the logistic regression model to group 1, a *p* value of 0.09 was calculated. Since this outcome is below the 0.3 threshold, it strongly suggests that the breach pike material was linked to a direct iron-making process.

### 3.4. Thermodynamic Analysis

The lower limit of the maximum temperature attained during the direct smelting process was roughly evaluated with a ternary phase diagram and Rhyolite–MELTS software (https://melts.ofm-research.org/ accessed on 26 December 2023). In particular, the average chemical compositions of groups 1 and 2 (Table 1) were projected into the FeO–SiO$_2$–CaO ternary system (Figure 16). Minor oxides were included in the analysis, pairing acidic oxides (Al$_2$O$_3$, TiO$_2$, P$_2$O$_5$) with SiO$_2$ and basic oxides (MgO, Na$_2$O, K$_2$O) with CaO, whereas MnO was combined with FeO. It was observed that group 1 was situated near the boundary between the tridymite and wollastonite primary fields, while group 2 fell within the wüstite primary field. Accordingly, the liquidus temperatures for groups 1 and 2 were approximately 1200 °C and 1300 °C, respectively (Figure 16).

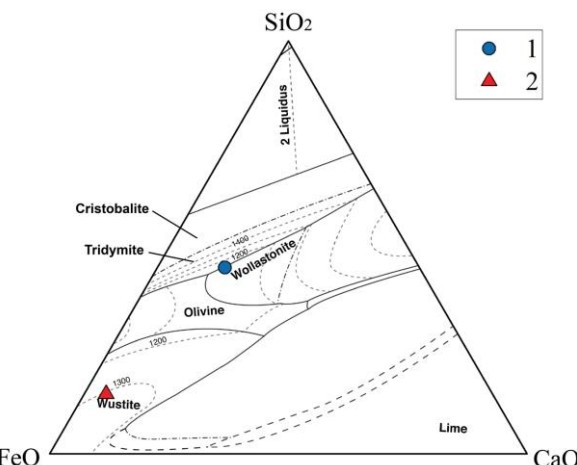

**Figure 16.** SI groups bulk chemistry projected onto the (FeO + MnO) – (SiO$_2$ + Al$_2$O$_3$ + TiO$_2$ + P$_2$O$_5$) – (CaO + MgO + Na$_2$O + K$_2$O) system.

The liquidus temperature corresponding to group 1 was also estimated by employing the Rhyolite-MELTS software, which is effective for modeling multicomponent oxide systems [24,25]. A thermodynamic computation was conducted based on the mean chemical composition of group 1 (Table 1). Additionally, a constant pressure of 1 bar (0.1 MPa) was set, whereas the oxygen fugacity was constrained to the iron–wüstite (IW) buffer (input files are provided in the Electronic Supplementary Material). The outcome for smelting-related group 1 was 1275 °C. This result can be considered an indicator of the minimum temperature peak reached during the smelting process.

To estimate the bloomery working conditions (i.e., the temperature and oxygen chemical potential of the reducing gas phase) and expand the SI classification in terms of disequilibrium levels, the thermodynamic-based strategy described in the Section 2 was adopted. The computed thermochemical parameters for each equilibrium-constrained SI–metal system are shown in Figure 17a. It was observed that the equilibrium temperatures exhibit a bell-shaped distribution in the relative frequency histogram, spanning a range from 816 °C to 1353 °C (Figure 17b).

Moreover, a central tendency was found at 1100 °C. The SI–metal systems that equilibrated between 1000 °C and 1200 °C were classified as low-disequilibrium systems (labelled "B" in Figure 17b). It should be emphasized that only for these systems did the modelled thermochemical parameters likely approximate real equilibrium conditions. Specifically, the values of the temperature and oxygen chemical potential for low-disequilibrium SI–metal systems positioned in the three main zones (F + F**, F + P, and P + F*) are summarized in Table 5. In particular, the equilibrium temperatures vary between 1095 °C and 1118 °C, whereas the oxygen chemical potentials ($\mu_{O2}$) range from −442 to −374 kJ/mol. These thermochemical parameters provide a reasonable approximation of the operating conditions for the bloomery smelting process. In addition, it was noted that low-disequilibrium systems are prominently observed in both the highest and lowest carburized zones (Figure 17c,d). In particular, the systems located in high- and low-C zones were associated with low and high values of oxygen chemical potential, respectively (Table 5). These features seem to exclude the occurrence of post-smelting carburization and decarburization, as these thermochemical treatments would greatly increase the degree of disequilibrium [19].

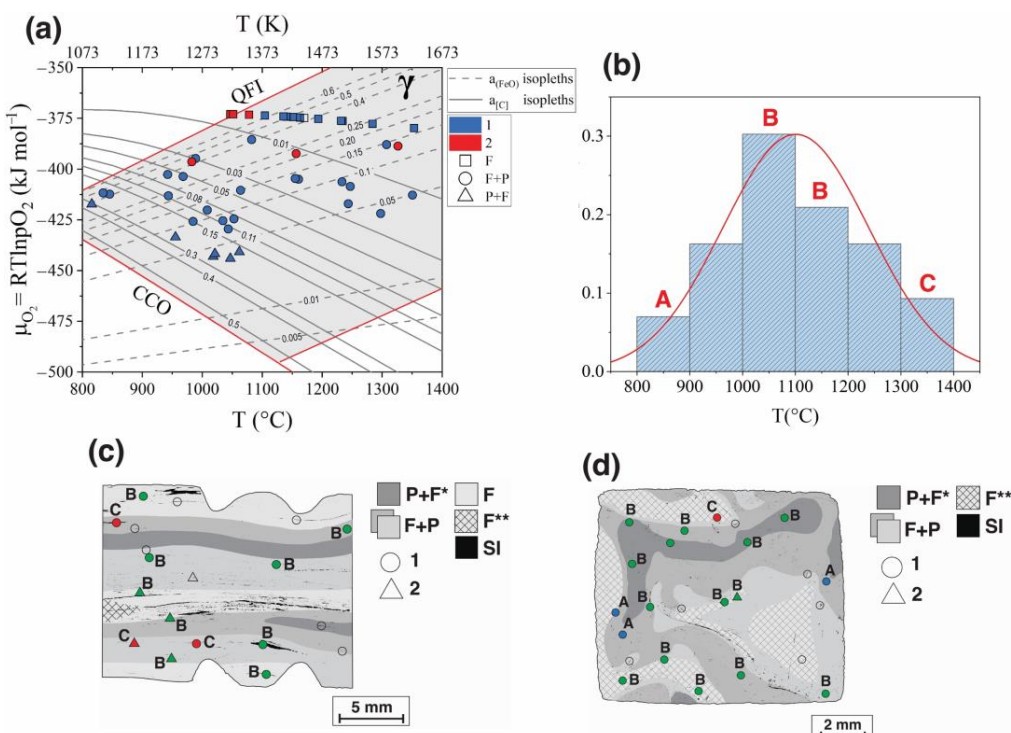

**Figure 17.** (**a**) Equilibrium conditions for the SI–metal systems. Quartz–fayalite–iron (QFI) and C–CO–CO$_2$ (CCO) buffers delimit the austenite stability field (grey-colored area). (**b**) Frequency distribution of estimated equilibrium temperatures (A, C = high disequilibrium, B = low disequilibrium). Position of high- and low-disequilibrium systems on microstructural constituent maps of (**c**) section S2 and (**d**) section S3 (F = idiomorphic and equiaxed ferrite, F* = idiomorphic, allotriomorphic, and Widmanstätten ferrite, F** = ferrite characterized by abnormal grain growth, P = pearlite, SI = slag inclusions; SI of types A, B, and C are represented in blue, green, and red colors, respectively).

**Table 5.** Values of equilibrium temperature (T) and oxygen chemical potential ($\mu_{O2}$) for low-disequilibrium SI–metal systems (F = idiomorphic and equiaxed ferrite, F* = idiomorphic, allotriomorphic, and Widmanstätten ferrite, F** = ferrite characterized by abnormal grain growth, P = pearlite).

| Metal Microstructure | T (°C) | $\mu_{O2}$ (kJ/mol) |
|:---:|:---:|:---:|
| F + F** | 1118 | −374 |
| F + P | 1084 | −411 |
| P + F* | 1095 | −442 |

Conversely, few SI–metal systems exhibit notable positive and negative deviations from the central tendency. These systems were likely characterized by a high disequilibrium level, and, therefore, the corresponding equilibrium-constrained results are probably not representative of real thermochemical conditions. In particular, SI–metal systems that equilibrate at a temperature above 1300 °C were interpreted as high-disequilibrium systems (labelled "C" in Figure 17b), typically featuring high-FeO SI surrounded by a high-C metal matrix (Figure 17c,d). In addition, SI–metal systems that equilibrated at temperatures below 900 °C were also classified as high-disequilibrium systems (labelled "A" in Figure 17b), predominantly consisting of low-FeO SI interfacing with a low-C metal matrix (Figure 17c,d).

## 4. Conclusions

This research work was dedicated to the archaeometallurgical investigation of a Renaissance breach pike. The aim of this study was to effectively reconstruct its manufacturing process in terms of the iron-making and forging technologies employed. For this purpose, a strategy based on metallography, Vickers microhardness tests, and slag inclusion analysis was adopted. In particular, the metallographic results and microhardness measurements point out that the breach pike was forged by hot hammering starting from a highly heterogeneous hypoeutectoid steel bar characterized by a significant amount of SI. Specifically, the conical socket was formed by first flattening one side of the lump and then wrapping it around a mandrel. Subsequently, the socket–spike intermediate zone was rounded by means of a proper swage block. In addition, the quadrangular spike was shaped by drawing out and pointing smithing operations. After forging, the geometry of the socket–spike transition zone was modified by carving two circular grooves. Moreover, the ferritic–pearlitic microstructures detected in sections S1, S2, and S3 taken from the socket, the socket–spike transition zone, and the middle spike, respectively, are consistent with iron/steel air cooling. The presence of a bainitic microstructure in section S4, which was sampled near the spike tip, can be explained by more rapid cooling, which may have been intentionally promoted to increase the tip hardness. The compositional data of a large sample of slag inclusions were collected by SEM/EDS microchemical analysis. To detect smelting-related SIs, the dataset was explored following a methodology based on multivariate statistics (PCA/HCA) and an analysis of non-reduced compounds. It was found that all the analyzed SI were likely derived from the smelting process. Moreover, the analysis of the SI chemical composition by a logistic regression model revealed that the breach pike raw iron/steel was likely produced through a direct iron-making method. The lower limit of the maximum temperature reached during the bloomery smelting process was estimated by computing the slag inclusion liquidus temperatures following two different approaches: a ternary phase diagram and Rhyolite–MELTS software. The analysis revealed that the minimum temperature peak was in the range of 1200 °C to 1300 °C. Finally, a thermodynamic-based model was used to expand the SI classification in terms of disequilibrium levels and compute the average thermochemical parameters (i.e., the temperature and oxygen chemical potential of the gas phase) of the bloomery smelting process. Focusing on low-disequilibrium systems, the computed temperature values vary from 1095 °C to 1118 °C, and the oxygen chemical potentials ($\mu_{O2}$) range between $-442$ and $-374$ kJ/mol. Additionally, the identification of low-disequilibrium slag inclusion–metal systems in all low- and high-carbon areas rules out the possibility of post-smelting carburization and decarburization phenomena.

**Supplementary Materials:** The following supporting information can be downloaded at: https://www.mdpi.com/article/10.3390/met14010041/s1.

**Author Contributions:** Conceptualization, methodology, software, formal analysis, P.M.; resources and investigation, P.M.; writing—original draft preparation, P.M.; writing—review and editing, M.F., G.C. and R.G.; supervision, M.F., G.C. and R.G.; project administration, M.F. and G.C. All authors have read and agreed to the published version of the manuscript.

**Funding:** This research received no external funding.

**Data Availability Statement:** Data are contained within the article and supplementary materials.

**Acknowledgments:** The authors are grateful to Leonardo Lauri for performing the SEM/EDS analyses.

**Conflicts of Interest:** The authors declare that they have no conflicts of interest.

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
