# Peer review of "Exploring the Manufacturing Process of a Renaissance Breach Pike"

_metals, doi:10.3390/met14010041_

Round 1
Reviewer 1 Report
Comments and Suggestions for Authors
Define the SI abbreviation the first time it is used on line 72.
It can be published in this form, but it would help readability if the thermodynamic analysis was described in more details.
Reviewer 2 Report
Comments and Suggestions for Authors
The contribution of M. Merico and his co-authors provides useful information for reconstructing the manufacturing process of a Renaissance breach pike, a type of object that has been the subject of very few archaeometallurgical studies up to now. The authors have taken particular care to qualify the ferrous alloys that make up the object, and to reconstruct the conditions under which the metal was reduced. The approach adopted is based on up-to-date, well-proven approaches, which are well contextualized through the proposed bibliography. They also employ methods they themselves have developed to study thermo-chemical processes applied to ancient alloys. Overall, the discourse is clear and well supported by good-quality illustrations. In addition, the article provides an original approach in terms of modeling slag formation conditions (particularly with regard to equilibrium conditions), in order to describe more precisely the chemical classes that are defined by multivariate means.
In my opinion, this contribution is of high quality and can be published in the special issue of ‘Metals’, but some modifications have to be made to this manuscript prior to publication, both to make up for a lack of contextualization of the object, but also to better support some proposed hypotheses. Corrections also need to be made to the multivariate approach adopted to describe slag inclusions.
The main revisions that I feel need to be made are as follows:
- Introduction: the context of the object has to be clarified; is there any information about the discovery? Is it from an archaeological excavation? Is it from a museum collection? If so, which one? What is its inventory number? Do we have more precise historical information on the collection from which it came?
- Line 99-101: Is the assessment method of carbon content based on fractions of distinct components of the alloy proven? It would be useful to cite a publication establishing that there is a strict correlation between carbon content and surfacic proportions of phases.
- Line 300-306: The bainite formation observed on the S4 section is explained by the fact that the metal cross-section is smaller than for S3, which does not contain these structures. However, the difference in cross-section (a ratio of 1 to 2 in the two dimensions) does not seem to me to be sufficiently significant to be satisfied with this explanation. The text suggests that the entire body of the pike has been homogeneously shaped. The possibility that the tip has undergone work sequences distinct from the rest of the object, by undergoing specific heatings, and therefore that particular attention has been paid by the blacksmith to this part of the object would also have to be discussed.
- Some revisions are required to part 3.2. SI classification. First, there is a population difference between the figure representing the factorial plans and the AHC presented in the Supplementray Materials on the one hand, and the location of inclusions on sections S2 and S3 on the other. On the latter, there are 8 inclusions from group 2, whereas only 6 inclusions are identified by the multivariate approach. Secondly, the symbology of markers on the factorial plane needs to be simplified, representing only solid markers. It seems that the current symbology is linked to a classification involving a lower level of inertia on figure S1 (between approx. 0.4 and 0.9), whereas a level of 2 was used for the classification.
- Lines 344-348 : the authors observe an anticorrelation between SiO2 and MnO for some of the inclusions classified as group 1. The explanation submitted (i.e. partial reduction of Mn) does not fit in my opinion to the rather strong trend revealed by the scatterplot. Manganese may be reduced to mixed Fe and Mn carbides starting from app. 1220°C (following Dillmann 1997, Diffraction X, Microdiffraction X et Microfluorescence X sous Rayonnement Synchrotron et analyses comparées pour la caractérisation des inclusions. Application à l'étude de l'évolution historique des procédés d'élaboration des objets ferreux (procédés direct et indirects), in Sciences Mécaniques pour l'Ingénieur. 1998, Universite de Technologie de Compiegne: Compiegne. p. 300) but in low quantities. Given the temperatures assessed by the authors for smelting (ranging 1200-1300°C, maybe a bit more), it seems unlikely that that much Mn would have been reduced. A way to explain this phenomenon could be the local formation of high-content Mn phases during smelting, whose uneven repartition within the slag would have led to concentration effects. A way to explore this hypothesis would be to 1) examine the mineralogy of SIs that display this phenomenon 2) locate these inclusions within the pike to see if they are evenly distributed within the metal, or if they are located in a given part of the object.
Some additional comments:
There is a seam on the polished section in figure 4. Could this correspond to the large line of inclusions seen in parts a and b of figure 5? If so, it would be interesting to mention it, as this would provide more information on how the object was made.
Line 90: change spelling to "Mitutoyo".
Line 110: sandy addition is mentioned, but this is not the only flux used in early forging. It would be preferable to remain more general, simply mentioning "fluxes".
Line 189: The term "poor quality" is problematic. It is a subjective assessment by present-day scientists examining old objects. The assessment of quality is the subject of debate in the archaeological and technical historian communities. Quality is first and foremost appreciated by the user, related to the properties that are expected of it in relation to its intended use. Quality can be assessed in a relative way, by comparison with a large series of similar objects, which cannot be the case here. I would remove this expression or find a more objective formula to describe alloys.
Reviewer 3 Report
Comments and Suggestions for Authors
The article is very interesting and well written. The SEM images are very good and illustrate the article well. I have several questions regarding the slag inclusions:
How large were the analysed slag inclusions? How many inclusions were discarded from the analysis? Were they analysed by area or point analyses?
How accurate are the analyses? They are given with two numbers after the point, this suggest a high accuracy.
Is the ore source known or suspected? High manganese ores are not that common, Austria? Noricum? Is it possible that the manganese originates from a manganese rich sand used in the smithing process?
In the conclusions it is mentioned that all slag inclusions are derived from the smelting phase but what about the inclusions from group 2? When are they formed?
It would be good to visualize the shaping of the pike in different phases. Is reworking (maybe after damage) visible in the metal structure?
